# How Can You Not Shout, Now That the Whispering Is Done? Accounts of the Enemy in US, Hmong, and Vietnamese Soldiers' Literary Reflections on the War

**David Beard**

Department of English, Linguistics, and Writing Studies, University of Minnesota Duluth, Duluth, MN 55812, USA; dbeard@d.umn.edu

**Abstract:** As typified in the Christmas Truce, soldiers commiserate as they see themselves in the enemy and experience empathy. Commiseration is the first step in breaking down the rhetorical construction of enemyship that acts upon soldiers and which prevents reconciliation and healing. This essay proceeds in three steps. We will identify first the diverse forms of enemyship held by the American, by the North Vietnamese, and by the Hmong soldiers, reading political discourse, poetry, and fiction to uncover the rhetorical constructions of the enemy. We will talk about both an American account and a North Vietnamese account of commiseration, when a soldier looks at the enemy with compassion rooted in identification. Commiseration is fleeting; reconciliation and healing must follow, and so finally, we will look at some of the moments of reconciliation, after the war, in which Vietnamese, Hmong and American soldiers (and their children and grandchildren) find healing.

**Keywords:** enemyship; rhetoric; reconciliation; commiseration; Vietnam/Vietnamese; Hmong; war

---

The critical reflections at the core of this essay begin with a conversation between a student and I, after my course (in multigenre writing through the Vietnam War) was over and the student had graduated, stopping just to say "hi".

The student enjoyed the class, they said, but they wanted to encourage me to talk about "the other side" of the war. I asked whether they had any suggestions. They talked about protest movements, Kent State, American popular music against the war.

It occurred to me, in that moment, that the student imagined that "both sides" of the Vietnam war included the hawks who sent our military into war and the hippies who protested it. Both sides of the war were inside the United States. As I had taught the class, and as our popular culture had reinforced within the student's mind, the Vietnamese were not, for this student, one of the "sides" of the war.

I'm not entirely sure that the Vietnam war is reducible to an experience of "sides". My student's intuition that there were two "sides" to the American experience of the war could be pulled further—maybe, within the U.S., there were more than two sides. Those who were "for" the war within the U.S supported it for diverse reasons; those who were against the war resisted it for diverse reasons, too. Maybe there were five or six "sides" to the war, even, within the United States.

In Vietnam, too, the war was built from more than two "sides"—the American soldiers who had been the center of my class entered the war with allies, and those allies were more than "on our side". The North Vietnamese had allies, too, invested in the war for more than one reason.

I went home to revise the class: the next iteration of the class complicated our sense of the "sides" of the war by more deeply addressing the North Vietnamese and Hmong soldiers who fought in the war. This essay, for this special issue of *Humanities* on war in literature, brings American, Hmong, and Vietnamese representations of the enemy into dialogue as we explore the themes of commiseration and reconciliation.

## 1. About Enemyship: The Rhetorical Construction of the Enemy Complicates Commiseration

Accounts of "enemyship" focus on the ways that political leaders use the construction of an enemy to bring unity to a (typically national) community. In *Enemyship: Democracy and Counter-Revolution in the Early Republic*", Jeremy Engels demonstrates the rhetorical process by which politicians construct "mutual antagonism for an enemy, resulting in a solidarity of fear, a community of spite, a kinship in arms, and a brotherhood of hatred" (Engels 2010, p. 13). To justify a war, the United States needed to project "fears outward onto an enemy who had to be eliminated" (Engels 2010, p. 20). If friendship brings a people together, enemyship unites them, too, in opposition to someone else, real or imagined.

Frank and Park connect enemyship to foundational myths; in examining the political discourses of Korea, they examine the ways that "foundational myths are formed, the trajectories they take, the enemyships they create" (Frank and Park 2018, p. 221). Given that enemyship is born and given structure by foundational myths, and that the foundational myths will be different across either side of the battlefield, we can conclude that enemyship in war is asymmetrical. In the Korean War, the Koreans were shooting at American soldiers for reasons different from the reasons that the Americans were shooting at the Koreans. Soldiers on one side believed that they were repelling an invader in what they now call the "Great Fatherland Liberation War". On the American side, soldiers were holding a Communist ideology at bay on foreign shores.

Such asymmetry is also evident in the Vietnam War. We call the conflict "the Vietnam War", while Vietnamese people refer to the same conflict as "the American War", implying a symmetry. However, there is no more symmetry in the Vietnam conflict than there was in the Korean conflict. The North Vietnamese construction of their enemy (the Americans) is not symmetrical to the American construction of their enemy, the Communist Vietnamese. And the asymmetry is complicated again by a third population, our allies in the Secret War, the Hmong. The Hmong construction of their enemy differs from both the American and Vietnamese sense of enemyship.

This essay draws our attention to literature which shows soldiers in commiseration in a time of war, despite the rhetorical and ideological forces driving them against each other. Commiseration, as a form of empathy, begins with seeing the enemy as similar to ourselves: *we are all soldiers here*. This is the first step in breaking down the rhetorical construction of enemyship that acts upon soldiers. After commiseration, soldiers can move toward reconciliation.

This essay proceeds in three steps. We will identify first, the diverse forms of enemyship held by the American, by the North Vietnamese, and by the Hmong soldiers, reading political discourse, poetry and fiction to uncover the rhetorical and ideological constructions of the enemy.[1] We will talk about both an American account and a North Vietnamese account of commiseration, when a soldier looks at the enemy with compassion rooted in identification. Commiseration is the first step toward healing. Finally, we will look at some of the moments of reconciliation, after the war, in which Vietnamese, Hmong and American soldiers (and their children and grandchildren) find healing.

---

[1] For these claims, we will look at poetry and literature both by poets and authors as well as by writings of soldiers. Of special note are texts from *Note on Captured Documents*, which are a unique set of poems. "American military units captured documents—among them a variety of military communications, as well as personal diaries, letters, and, surprisingly, a great deal of poetry written by members of the various revolutionary forces. These were sent to Saigon for analysis and translation, grouped into batches, and microfilmed on 35-mm film stock. (The originals were lost in 1975.) The microfilm copies were housed in the National Archives in Washington, D.C., and after the war they eventually were declassified" (vii–viii). The collection is nineteen miles of microfilm. The volume of poetry published from this microfilm is especially useful for clarifying a soldier's perspective. "The effect of this poetry is to humanize the soldiers who fought on the side of the revolution in such a way as to help dispel the stereotypes created by the United States military and the American media during and after the American war in Vietnam" (p. xiii).

*1.1. The American Sense of Enemyship, 1955–1975*

While some American soldiers entered Vietnam with a sense of nationalism and heroism, many were reluctant draftees. American construction of the enemy in the North Vietnamese was energized by two forces: by political ideology and by racist experience. We will discuss each in turn.

American political ideology constructed the North Vietnamese as a proxy for Communism. The North Vietnamese had not invaded the United States, nor did they intend to do so, but Communism was a global threat, manifest in the "domino" theory. The threat posed by North Vietnam was indirect and a construct of Cold War propaganda.

The Gulf of Tonkin Resolution becomes a moment in which the U.S. government could argue that the North Vietnamese were no longer only an ideological threat. As Richard Cherwitz demonstrated (in a series of articles on the Gulf of Tonkin resolution[2]), Johnson's administration took the opportunity of the crisis in the Gulf of Tonkin to remake an ideological opponent into a physical threat.

In terms of enemyship, then, the American soldiers were, in a way, shooting at Communism by shooting at the North Vietnamese. But time in country would wear on the soldiers, and they would eventually not only shoot at North Vietnamese; they would shoot at all Vietnamese. Racism became part of the experience of enemyship of American soldiers.

The lived experience of a guerilla war disoriented American soldiers; they ceased firing only upon the Viet Cong and simply began shooting the Vietnamese. As Jean Paul Sartre noted in his work against the Vietnam war, "The armed forces of the United States torture and kill men, women and children in Vietnam because they are Vietnamese" (Sartre 1968, p. 82). The guerilla nature of the war, in the end, eroded the ability of the American soldiers to determine which Vietnamese people were friend or foe.

Sartre gives a rich and violent description of the change in the behaviors of U.S. soldiers:

> (T)he racialism of the Yankee soldiers from Saigon to the 17th parallel has increased. The young Americans torture without repugnance, shooting at unarmed women for the pleasure of completing a hat-trick: they kick the wounded Vietnamese in the testicles; they cut off the ears of the dead for trophies. The officers are worst: a general was boasting in front of a Frenchman who testified at the Tribunal of hunting the Viet Cong from his helicopter and shooting them down in the rice fields. They were, of course, not NLF fighters, who know how to protect themselves, but peasants working in their rice fields. In these confused American minds, the Viet Cong and the Vietnamese tend to become more and more indistinguishable. (Sartre 1968, p. 81)

Sartre hammers the description home: "A common saying is 'The only good Vietnamese is a dead one,' or, what comes to the same thing, 'Every dead Vietnamese is a Viet Cong'" (Sartre 1968, p. 81). At some point, for some soldiers, they ceased shooting at specific Vietnamese as proxies for Communism and started shooting at anyone who was not visibly American.

Sartre's account has been borne out by historical research. As Turse (2013) explores in *Kill Anything That Moves: The Real American War in Vietnam*, the widespread deaths of civilians at the hands of US soldiers in Vietnam were no accident. Turse counts 65,000 North Vietnamese and 3.8 million South Vietnamese civilians dead by the end of the war—some were killed by "friendly fire", but many were simply murdered. Col. David Hackworth was quoted (by Patricia Sullivan (2009) in the *Washington Post*) as claiming that "My division in the Delta, the 9th, reported killing more than 20,000 Viet Cong in 1968 and 1969, yet less than 2000 weapons were found on the 'enemy' dead. How much of the 'body count' consisted of civilians?" In different ways, the demands for high body counts (for purposes of creating a narrative of American success to deliver to the media back home) gave soldiers permission

---

[2] Cherwitz (1978, 1980) makes these arguments in "Masking Inconsistency: The Tonkin Gulf Crisis" and in "Lyndon Johnson and the 'Crisis' of Tonkin Gulf: A President's Justification of War".

to shoot any Vietnamese. This policy converged with the psychology of the individual soldier (feeling under threat from every direction) to encourage the murder of civilians. The military and the media incentivized what fear accelerated: a desire to kill anyone who wasn't obviously an American.

All of that said, the experiences of soldiers are diverse, and this short essay cannot be exhaustive. (For example, enemyship was not always central to the solder's experience; for some soldiers, enemyship mattered less than staying alive long enough to come home.) In terms of the rhetorical processes of enemyship, though: for two decades, the enemies of many American soldiers of the Vietnam war were "Communism", or "the Vietnamese", as a racial category.

*1.2. The North Vietnamese Sense of Enemyship, 1887–1975*

The sense of enemyship among the North Vietnamese soldiers was radically different from the sense of enemyship among the American soldiers. We will point to two salient features of the rhetorical and ideological construction of the enemy among the North Vietnamese soldiers. First, the North Vietnamese understood the war as the next phase in a century-long war against colonial powers (of which the United States was just one). Second, the Vietnamese understood the war as a war to repel invaders, and their work as soldiers was to protect their nation, envisioned as their lover, from these invading forces. We will discuss each in turn.

*First:* for North Vietnamese soldiers, the conflict did not begin with the arrival of U.S. advisors in 1955 or with formal U.S. military participation in the war in 1964. Rather, in the lived experience of some of its soldiers, the beginning of the war dated to 1946 (the first "Indochina war" against France). For others, the war began with the 1887 formal recognition of Vietnam as "French Indochina" (the moment when the Vietnamese lost control over their home). Soldier and poet Duc Thanh tells us, in "In the Forest at Night", that he is "the son of the Vietnamese, under siege for a hundred years By the French and Americans" (Nguyen and Weigl 1994, p. 45). For Duc Thanh, the war is not a war against the United States. It is not a war to advance Communism (as American propaganda claimed). Duc Thanh fought a war against colonial powers.

Huang Loc names his colonial enemies with rancor in "Condolence to a Friend". Writing from the Quang Tri campaign in 1972, he asks "Who took that fatal shot? What gun hit the mark? Please, sacred spirit, show me The murderer, call out his name". The answer is cold and hard, and it makes clear the Vietnamese experience of the war: "He's an imperialist. He's a colonialist. He's a bandit" (Bowen 1998, p. 47). The enemy is not an American, a Yankee who must go home. The American War, in the minds of its soldiers, was just one more manifestation of the century-long fight for freedom from the colonizers.

*Second:* the enemy, in the mythic narrative of the Vietnamese soldiers, comes to attack the homeland whom the soldiers must defend as they would defend a lover. Nguyen Dinh Thi writes a poem addressed to a lover back home, using that form of address to explain his experience of war. In "Remembering", he notes, "I love you as I love our country, In pain, hardship and with great passion. Every step I take you are in my thoughts, Every meal I eat, every night I sleep" (Bowen 1998, p. 15). That passion is with him every day; it fuels his efforts in war. This love of country, imagined as a romantic love, is powerful: "We'll fight all our lives for our love. We love each other, and we are proud to be human" (Bowen 1998, p. 15). The impulse to defend a lover against an invader is deeply human.

As among the Americans, motivations for fighting in war are diverse; I offer here only a sample. At a rhetorical level, some of the North Vietnamese soldiers fought the war to protect their nation (envisioned as their lover) from colonialism.

### 1.3. The Hmong Sense of Enemyship, 1910–1975

The Secret War, in Laos, complicates the diverse experience of enemyship in the Vietnam War. The Hmong brought their own stories to their experience of enemyship. During the Hmong struggle for self-determination against a backdrop of shifting colonial powers, the nature of the enemy shifted.[3]

The Hmong had rebelled against the French colonial powers from 1910 to 1912 (in the conflict called "Mi Chang's rebellion"), and from 1918 to 1921 (called the "War of the Insane" by the French). Both rebellions were suppressed by the French. In World War II, the French shifted from enemy colonizer to ally, as the Hmong sided with the French to push Japanese occupiers out.

After WWII, between 1946 and 1954, Hmong led by Touby Lyfoung resisted the North Vietnamese, while Hmong led by Faydang Lobliayao resisted French colonial rule. In both cases, the Hmong were fundamentally seeking autonomy. Touby Lyfoung believed that autonomy was better preserved by alliance with the French against the Communist North Vietnamese; Faydang Lobliayao believed that autonomy was better preserved by alliance with the North Vietnamese against the French.

When Vang Pao formed alliances with the United States, he was, like Lyfoung, allying with the western colonial power against the Communist colonial power. In return, the U.S. airlifted roughly 40 tons of food per month to Hmong communities and funded new schools throughout the remote regions of Laos, training Hmong girls as nurses and medics to care for wounded soldiers and increasing Hmong literacy.

While Americans and North Vietnamese were "taking sides" against each other, the Hmong found that the war came to them. Lao poet Bryan Thao Worr's claim that "There were refugees Who to this day cannot explain why they were the enemy When the war came" (Moua 2002, p. 98) makes sense when we see the Hmong as participants in someone else's war narrative, rather than protagonists of their own war story. Hmong served as allies of the United States in fighting the Secret War, but they were, in many ways, fighting a different war, than their American allies fought.

On the one hand, some Hmong saw the Americans as allies against Communism: Mai Der Vang asks "Do you think of your missing wife, how the Pathet Lao dragged her naked, screaming, and bleeding by her long black hair deep into forest shadows" (Der Vang 2017, p. 7). Pathet Lao were a communist political movement in Laos; the Hmong were allies of the U.S. against Pathet Lao, and so against Communism, in the Secret War.

> And yet, there was little affection for the Americans: Mai Der Vang asks
>
> Do you think of the American
>
> returning to the coffee cup
>
> new linens
>
> in a warm bed
>
> pulling into the driveway? (Der Vang 2017, p. 7).

The Hmong sacrifices in the Vietnam war dwarfed American sacrifices. By war's end in 1975, nearly 40,000 Hmong soldiers had been killed or were missing in action—totaling approximately one-fourth of all Hmong men and boys. For point of comparison, the US death total in Vietnam was about 58,000 when the total US population was 200,000,000. Hmong soldiers, many of whom were children when they were asked to bear arms, resented the Americans who could go home to comfortable lives.

Bryan Thao Worr tell us that, in retrospect, other soldiers (not the Viet Cong, nor even the Americans who left them behind) were not the real enemy of the Hmong. Instead, the Hmong people "are victims of fat tigers and foreign policy" (Moua 2002, p. 98). In this way, Hmong experience of the war and of enemyship was their own.

---

[3] This history is traced in broad strokes from (Hillmer 2010; Her and Buley-Meissner 2012).

At this point, I'd like to pause and note the ways that the rhetorical study of enemyship changes our understanding of war. War is often metaphorically understood in terms of games. Board games have for centuries appeared to model warfare. But when two players meet to play chess, they have symmetrical objectives in the same system. Player push their pawns across the board hoping to achieve checkmate, victory on the battlefield. When paintball players meet to wargame, each team seeks to capture the flag of their opponent's team. But the perspective of enemyship reminds us that real war doesn't work that way. War is asymmetrical, even chaotic, in this light.

A slow walk through the distinctions between American, North Vietnamese, and Hmong rhetorical constructions of enemyship demonstrates how impoverished this "two sides" conception of war is. In the next section, I will map the implications of the study of "enemyship" for the possibility of commiseration on the battlefield.

## 2. Commiseration on the Battlefield

I have sketched here the ways that soldiers from three communities envisioned their enemies in war. It might feel like these soldiers were fighting three different wars, where the only common ground was the battlefield. On the battlefield, though, commiseration was possible.

In the broader history of warfare, the prototypical moment of commiseration is the Christmas Truce of World War I, in which "soldiers began to brave No Man's Land, shaking hands, exchanging gifts and pictures, mutually complaining about the war, and kicking a ball around" (McDaniel 2015, p. 92). In these moments, the soldiers on both the German and British sides of the front paused and saw themselves in the enemy. They both celebrated, even played, with each other—but they also used this time to "repair their trenches as well as collect and bury the dead"—very, very human acts made possible because the Germans and British soldiers saw past the rhetorical construction of the enemy. Instead, they saw in each other their common Christianity and their common exhaustion as soldiers. They looked across No Man's Land and saw people like themselves (instead of enemies).

Given all that we have said about enemyship in the Vietnam war, it's not surprising that there could be no collective moment like the Christmas truce. There is no common identity, like religion, that binds North Vietnamese, Hmong, and American soldiers. But there are individual moments of commiseration across individual soldiers. To see those moments, we will look briefly at poetry by Nguyen Duy, who served in the North Vietnamese National Guard, and at a seminal short story by Tim O'Brien from the classic novel, *The Things They Carried*.

In "Stop", Nguyen Duy talks about the "ranger with the face of a child" whose "shot just missed" him. He's very aware that as the soldier flees, Duy has the upper hand: "If my finger moved half a millimeter, he'd be dead" (Bowen 1998, p. 133). But doesn't want to press that advantage. After all, "if the situation had been reversed, and I ran in front empty-handed, and he ran behind, M16 in hand, very likely I'd have died". Duy reaches a moment of commiseration as he becomes able to see himself in the American soldier.

Tim O'Brien also experiences commiseration through identifying with the enemy. In "The Man I Killed", O'Brien identifies with the North Vietnamese soldier whom he encountered on a trail and whom he killed with a grenade. He finds that man was "a scholar, maybe", resonating with O'Brien's narrator's identity as a student. He asserts that the man he killed was "born, maybe, in 1946", the same year the author was born, and that "he liked books" (O'Brien 2009, p. 118), like O'Brien.

O'Brien's narrator projects his psyche onto the man he killed: "the young man would not have wanted to be a soldier and in his heart would have feared performing badly in battle . . . He had no stomach for violence" (O'Brien 2009, p. 119) We know these to be traits of O'Brien's narrator, who also "was afraid of disgracing himself, and therefore his family and village" (O'Brien 2009, p. 119). In earlier chapters of the novel, it is clearly established that Tim does not want to go to war; rather, he is afraid of being thought a coward.

When Tim O'Brien's narrator writes that the man he killed "hoped the Americans would go away" (O'Brien 2009, p. 119), he likely speaks truth about the North Vietnamese soldier. He also projects his own hopes onto the enemy. Neither one of them wanted the Americans to be there.

In looking at the enemy and seeing himself, O'Brien does the same work that Nguyen Duy does, placing himself in the place of the other. The soldiers reject the rhetorical and ideological construction of the enemy and understand that their enemy struggles as they do. And so they experience commiseration.

Seeing yourself in someone else is the first step toward commiseration. Commiseration may not be enough, though. In the Christmas Truce, "parts of the trench line did continue to shell each other on Christmas, and [ . . . ] soldiers died on that day" (McDaniel 2015, p. 92), while after the Truce, "propaganda touting the barbarity of the enemy increased in quantity and viciousness" (McDaniel 2015, p. 92). Commiseration is ephemeral, and violence remains. Both Duy and O'Brien tell stories of killing their enemy despite the experience of commiseration. We need more; beyond commiseration, we need reconciliation and healing.

## 3. After the War, Healing at the Memorials to the War

Commiseration can occur, however fleetingly, on the battlefield, but reconciliation cannot occur in the moment of war. Reconciliation comes after, but it does not come easily. The scars of war run deep after the battle stops, as Hmong, American and Vietnamese poets alike have felt their lives and land damaged long after the war.

Among Hmong poets, in "Declassified", Mai Der Vang tells us about the "vividness of saffron and canary arriving as small showers". Vang references claims that the USSR sprayed Vietnam and Laos with chemical weapons in 1981—long after the official end of hostilities, visible as a kind of yellow rain that dried as a yellow powder (Der Vang 2018).[4] Abandoned munitions remind Bryan Thao Worr of war: "Grenade fishing in the aftermath of Phou Pha Thi Has lost its novelty To the man with a bullet fragment rattling In his body, slowly tearing him apart". (Moua 2002, p. 99).

Among Vietnamese poets, bones speak of the wounds not healed: Van Le's meditation on "Quang Tri", a provincial capital that suffered attack in the 1968 Tet Offensive, opens with the observation that "everywhere we dug there were white bones . . . What kind of foundation would they make for our house? "Were they our bones or their bones?" Van Le answers: "There are no American bones here. The Americans left years ago and took their bones with them. These skeletons, scattered all over our land, Belong only to Vietnamese" (Bowen 1998, p. 121).

Healing comes hard when the ground beneath your feet reopens the wounds, burns and blisters, and more than a half century later, the pain is still coming to light. Mai Der Vang exhorts: "May this secret war . . . burn and blister under its own nakedness", as it comes into view for American, Hmong, and Vietnamese alike (Der Vang 2018). Reconciliation seems unlikely when so many hurts have yet to heal.

And yet: pilgrimage has made healing possible. The Vietnam Memorial in Washington, D.C. is, first and foremost, the site of postwar reconciliation between American soldiers in the Vietnam War and the American public. The site houses multiple works of sculpture: [a] the abstract Maya Lin memorial wall that captivated the American imagination in 1982, [b] the "Three Servicemen" statues sculpted by Frederick Hart in 1984, honoring the veterans from the branches of the military active in Vietnam, and [c] the Glenna Goodacre "Women's Memorial", honoring the women who served in the war, dedicated in 1993. As the site incorporates additional statuary, more Americans experience healing and resolution in the Memorial grounds.

---

4    Some scientists insisted that the yellow rains were bee feces, dropped by swarms moving through the jungle. Six years after the fall of Saigon, the lens of the war still frames discussions of yellow pollen appearing on tree leaves; it is hard to move forward when the landscape reminds you of war.

Bobbie Ann Mason's novel *In Country* offers an account of the struggles of those born after the war to understand the experiences of their veteran fathers and uncles. Visiting the memorial is the crescendo of understanding the war. In the novel, the protagonist, young Sam, touches her father's name, which she shares. She "touches her own name. How odd it feels" (Mason 1986, p. 245). Similarly, in "Facing It" by Yusef Komunyakaa, the poet goes "down the 58,022 names/half-expecting to find my own in letters like smoke" (Komunyakaa 2001, p. 234). Citizens who have never been to Vietnam feel connected to the veterans who lost their lives.

The wall, and the healing that the wall makes possible, is experienced by soldiers, by citizens who stayed home, and by the generations born after the war's end. As Mason writes it, "How odd it feels, as though all the names in America have been used to decorate this wall" (Mason 1986, p. 245)—all Americans can feel reconciliation at the wall.

Perhaps more strikingly, North Vietnamese soldiers also feel healing in visiting the Wall. Nguyen Duy, who served for a decade as a member of the North Vietnamese National Guard, visits the Memorial in a poem filled with religious imagery, noting that "heaven's eyes stare" as "a Wailing Wall sings cries of the dead". At the wall, he feels "a time of chaos, sun and moon trading places" (Duy 1999, p. 227). We expect American veterans to experience healing and reconciliation at the wall, but that a Vietnamese soldier experiences the same may surprise us.

Duy finds that the rhetorically constructed narratives of enemyship slip away when he visits the wall. Instead, "this grief deepens, this agony endures". Reconciliation comes at the wall because Duy recognizes that the war was "a game for some, heartache for all" (Duy 1999, p. 227), American and Vietnamese alike. Enemyship dissolves, and a healing deeper than the typically fleeting experience of commiseration.

## 4. Healing Still Ahead

I wrote this essay as part of the project of deepening my class in multigenre writings about Vietnam.[5] In revising my class, I wanted to incorporate the Hmong experience to complicate the accounts of American and Vietnamese soldiers. In writing this essay, I have a final goal: to exhort us to remember that there is work ahead in achieving reconciliation. I live and teach in Minnesota, near one of the largest Hmong communities in the world—perhaps that experience more than anything reminds me of the healing ahead.

The Hmong experience of the Vietnam war is marginalized. The war was called "the Secret War", and for many, it is still a secret. The complex on the National Mall (that includes the Maya Lin, Frederick Hart, and Glenna Goodacre sculptures) honors U.S. Veterans, but it does not honor or even acknowledge our Hmong allies.[6]

Several local memorials to Hmong participation in the war do exist in the cities where Hmong refugees have settled. For example, the *Lao, Hmong and American Veterans Memorial* has been built in Sheboygan, WI. A community of six thousand Hmong (in a city with a population of fifty thousand) worked with the City of Sheboygan to produce the memorial. Similar memorials exist in Wausau, WI (population 40,000, with about 10% of that population Hmong) and in St. Paul, MN, whose Hmong population of 30,000 people is roughly ten percent of the city's population. These memorials are striking and powerful but still marginal in the American process for healing from the war.

---

[5] The project of more deeply internationalizing my courses has been supported by the Global Programs and Strategy Alliance of the University of Minnesota, whose "Internationalizing the Curriculum" has supported my work. Visit https://global.umn.edu/icc/about/index.html for more information.

[6] Hmong veterans have been recognized, in the *Laos and Hmong Memorial*, or *Lao Veterans of America Monument*, which rests in Arlington National Cemetery—geographically distant from the Mall. This monument is primarily visited by veterans and their families and so is not part of most visitors' experience of the war in Washington, D.C.

Millions of Americans and North Vietnamese alike will not experience reconciliation or healing with Hmong participants in the Secret War until the Secret War becomes part of the public, national, memorial discourse on the war.[7]

After all, we call Hmong participation in the Vietnam War "the Secret War", but it wasn't long ago that the Vietnam War was our national secret, the war we didn't want to talk about, at all. But we have come to speak of the war, we have found our tongue, and we must keep speaking until reconciliation is possible for all. Hmong poet Bryan Thao Worr tells us in "The Last War Poem" that we must speak:

> How can you not have words for the war of whispers?
>
> How can you not shout, now that the whispering is done?
>
> And I swear, each time I break this promise, that the next time
>
> Will be the last word I write about this damn war. (Moua 2002, p. 98)

The closure we need to enable Bryan Thao Worr to keep his promise will come from breaking down the rhetorical construction of enemyship. We will stop seeing the North Vietnamese as enemies, the Hmong as the allies we refuse to acknowledge. Beginning with what this special issue of *Humanities* calls "commiseration", we can move into reconciliation where, perhaps, we all can heal.

**Funding:** This research received no external funding.

**Conflicts of Interest:** The author declares no conflict of interest.

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
