# Peer review of "How Can You Not Shout, Now That the Whispering Is Done? Accounts of the Enemy in US, Hmong, and Vietnamese Soldiers’ Literary Reflections on the War"

_humanities, doi:10.3390/h8040172_

Round 1
Reviewer 1 Report
Thank you for the opportunity to read this essay. This is a very worthy topic—the impulse to look at rhetorical constructions of the enemy in the Vietnam War from a multisided point of view is a good one, and the idea of an asymmetrical viewpoint is well-taken. The author also gathers some fascinating literatures to the table, and the quotes are all well-chosen. The pedagogical connections are interesting in the introduction and conclusion. The chief issue, I think, in this draft is that there isn’t enough analytical work here to tie everything together. In other words, we have three movements here: enemyship, commiseration, and reconciliation—and those three concepts are not always defined well enough (or backed up by enough scholarship), and the author is often more describing the literatures rather than analyzing them and making a forceful argument. What does it mean to bring these three concepts together? What is the so-what? What do we take away that we didn’t already suspect before (the idea that war is multisided is important, but is that enough “news”)? I do understand that this is a short essay for a special issue, so a ton of depth might be difficult, but I do think the author can do more at the beginnings and ends of sections to advance an argument and to bring everything together into conversation.
Below are some more specific thoughts and areas that could use more attention in a revision:
- The writing is clear and jargon-free overall, which is great, but at times almost too simplified. On page 2, the author writes, ““Americans disliked Nazis for different reasons than Nazis disliked Americans, obviously.” This seems too informal and too surface-level to really register as a needed statement here. I would also look for other places where this happens.
- On page 3, the author says, “American construction of the enemy in the North Vietnamese was energized by political ideology and by racist experience.” That may be true, but there doesn’t seem enough depth or support here to make that very sweeping claim.
- Similarly, the paragraph at the top of page 4 feels like a simplification for the many reasons that the Vietnam War was fought. Some more scholarship to back this up might help.
- I worry that the Sartre quotes as the only evidence of America’s viewpoint on the enemy is not quite right. Why are we not getting American voices here to supplement that, like the North Vietnamese and Hmong get?
- Here is another spot where the writing gets too vague (on page 4): “At some point, for some soldiers, they ceased shooting at specific Vietnamese as proxies for Communism and started shooting at anyone who was not visibly American.” Again, this seems a broad and general statement that needs more support or more nuance.
- The first paragraph on page 5 needs scholarly support—how do we know that the North Vietnamese did not understand the war as a war against the US? Again, it’s not that the author is wrong, but these statements need more back-up.
- At the bottom of page 5: bottom could use better transitions between sections. Right now, the author is simply summing up the section, when this could be a real opportunity to advance the argument and connect to the next section.
- On page 6: where is the historical scholarship on this material about the Hmong? Let’s try and back up this historical material with scholarly support.
- On page 7, the author writes, “In this way, Hmong experience of the war and of enemyship was their own.” Again, this is true, but this seems a key spot to go into more depth. Here is where the author might bring all three versions of enemyship into conversation with one another and really analyze what it does for us.
- Similarly, the commiseration section doesn’t do enough to really analyze the process; great literature examples are used, but a more critical lens could be used by the author to tie it all together.
- I’m not sure reconciliation is defined in contrast to commiseration enough. How are we situating reconciliation here?
- And is the Vietnam Memorial truly a site of reconciliation? Isn’t it more complicated? How does this affect enemyship? These strands need tying together—the author does get to it a little bit toward the end (understanding enemyship can bring about commiseration and reconciliation), but it seems to need more depth.
- On page 9, this sentence needs more unpacking: “We expect American veterans to experience this energy, but that a Vietnamese soldier experiences this energy may surprise us.” This again may be true, but why does that surprise matter? What is the big takeaway here? This seems to be a crucial section where everything needs to be fully tied together.
- Watch out for typos—there are quite a few for a short piece, and should be worked out in the revision stage.
Author Response
Editors,
In regards to the required revisions to the enclosed submission to Humanities, I note that the editorial team made generous suggestions, easy to follow, which deeply improved my paper. I note the ways I responded to editorial comments below.
Corrections to Enemyship Section:
In response to one reviewer, I have clarified the section “About Enemyship” (lines 58-64) by following the reviewer’s directive to stay with an example from the Korean War. Given the analogies with the Vietnam war, this makes sense.
Corrections to American and Vietnamese Enemyship Sections
I enhanced the discussion of American motivations for entering the war by reference to the scholarship of Richard Cherwitz, whose work explains the tensions between an ideological construction of the enemy prior to 1964 and the construction of the enemy as a real, physical threat after the Gulf of Tonkin resolution. After fleshing out the section on political ideology, I supplemented my quotation from Sartre, using added evidence from Kill Anything That Moves: the Real American War in Vietnam, Turse, Nick, as recommended by one reviewer, and by reference to a first-person account from an American soldier, as recommended by the other reviewer.
TBH, I knew this section was weak, and I am grateful for the opportunity to flesh it out.
I made minor corrections at the sentence level throughout.
Corrections to Hmong Enemyship Section
I have added reference to the secondary sources that undergird the history of the Hmong. I also paused to do some synthetic work, explaining that reconceiving of war in terms of enemyship makes clear that board game or sports models for understanding war are inadequate to the task of describing the Vietnam war. In fact, I think, the notion of enemyship forces us to think of war through new or mixed metaphors. And, to be clear, it muddies notions of commiseration.
Corrections to Commiseration Section
Here, I did two kinds of work:
I defined commiseration by reference to the Christmas Truce, then noted the limitations of this model for application to Vietnam. There are no shared loci of identity among Vietnamese, Hmong and American soldiers to enable the kind of identification that British and German Christians shared; commiseration in that context is likely to be individual, rather than collective. I also worked to differentiate commiseration from reconciliation. Commiseration happens in war, is fleeting and notably, in its wake, violence still follows. Reconciliation happens after war, is lasting, and is healing.
Corrections to Reconciliation and Concluding Sections
The general fixes for the final two sections of the paper basically revolve around slowing down, explicating my ideas more clearly, explaining the healing function of the Wall more clearly – I didn’t really add more material here, but I let my ideas breathe more, with better explication.
Writerly Changes
I made several changes to the argumentative structure of the piece, more effectively signaling argumentative moves, providing a clearer overview of the work of the paper at the beginning, and tightening some chatty sentences, as recommended by the reviewer. I made minor corrections at the sentence level throughout.
Changes Not Made
I have resisted a few changes, but defer to the journal for style: For example, a reviewer indicated that “Line 30 reads better if “student” is changed to “students” since all the following references are plural ‘they.’” I intend a gender-neutral “they” there, as I am being intentionally ambiguous about the gender of the student. I could switch to “he” if the editor prefers.
Thank you for your kind attention.
Reviewer 2 Report
How Can You Not Shout, Now That the Whispering Is Done? Accounts of the Enemy in U.S., Hmong, and Vietnamese Soldiers’ Literary Reflections on the War
This article is an illuminating comparison of how U.S., Hmong, and North Vietnamese soldiers’ war literature view one another at these different stages: enemyship, identification, commiseration, reconciliation, and potential healing. The article analyzes various poems and stories to demonstrate how each culture progresses through those stages offering insight into the particular points of view. The concluding section offers commentary especially concerning the Hmong community.
The article is well organized, clearly defining each of the stages with succinct titles, well-developed themes and compact literary analysis. Included in that literary analysis is fresh research into a 2018 publication of declassified North Vietnamese soldiers’ poetry, which should be interesting to many war literature scholars. The comparisons are balanced and structurally grouped helping the reader to understand the various differences between the cultural viewpoints.
Overall, this article is particularly valuable in demonstrating what most combat veterans ultimately learn. Namely, that beneath the surface animosity portrayed in most war literature, the soldier may subtly realize the basic humanity of his enemy. That identification can then lead to commiseration, and perhaps, healing. The article points out that healing may only begin to occur away from the battlefield at an appropriate monument where introspection takes place.
This article will be a valuable addition to the ongoing discussions of Vietnam War literature.
There a few sections in the article that may benefit from a deeper scrutiny.
In the section “About Enemyship” (lines 58-64) the literature examined is about the Korean War, but the concluding example jumps back to WW II and the Nazis. Perhaps it might be better to stay with an example from the Korean War.
In the section “The American Sense of Enemyship” (lines 123-135) the author uses a strong quote from Sartre. As added evidence the author might consider drafting a footnote to include a recent publication verifying Sartre’s statement. Kill Anything That Moves: the Real American War in Vietnam, Turse, Nick. Metropolitan Books, 2013.
In the section “The North Vietnamese Sense of Enemyship” (lines 155-156) the author describes the soldier looking at the nation as a lover; however, this is not clearly defined or examined until line 176. Perhaps this idea might be defined in the earlier lines.
In line 158 the U.S. participation in the Vietnam War is dated 1955. This may draw some attention since the active American military involvement didn’t officially occur until 1964. Perhaps a footnote here would clarify that it was the American CIA that was actively involved in the 1950’s.
Overall, the article is well written, but a few editorial changes might be necessary
In line 28 “. . . between me and a student. . .” might be revised to read “between a student and me” or “between a student and myself.”
Line 30 reads better if “student” is changed to “students” since all the following references are plural “they.”
Line 72 seems somewhat awkward. Consider this revision. “Such asymmetry is also evident in the Vietnam War, which is the subject of this essay.”
Line 108. “Political Ideology” should be “American political ideology.”
Lines 143-145. This sentence seems awkward and the colon is misused.
Line 194. Change “. . . Vietnam is fluid, as the . . .” to “Vietnam is fluid during the . . .”
Line 247. Change “with” to “where”
Line 282. Change “z” to “the”
Line 339. Add the word “North” before “Vietnamese.”
Author Response

(The authors gave the same response as above.)

Round 2
Reviewer 1 Report
Thank you again for the opportunity to re-engage with the essay. I believe the author has largely made the necessary revisions to the piece, and with that effort, the essay is much improved and, I think, ready for publication. In a perfect world, the author might make one more revision where they continue to work on the writing/presentation a bit more. For example, it may be a matter of style, but I was not a fan of the new addition on the game/sports metaphor about enemyship--I think it might be too reductive for such a complex topic. I also am not sure that reconciliation and commiseration have been fully, well, reconciled :). In addition, I think some more textual editing might tighten everything up. But, in general, this is good work that I look forward to seeing in print.
Author Response
Thank you for this feedback.
I am willing to submit to the copyeditor's revisions, in terms of style. I know that copyeditors have a sense of style that I often do not.
Should I revisit the game metaphor? Maybe. I'd welcome suggestions from the editors.
As for the distinction between commiseration and reconciliation, yeah, the ephemerality of commiseration is the real distinction for me -- I can commiserate and then shoot, as the two examples I use demonstrate. Reconciliation is hopefully lasting.
I am open to revision directions from the editorial staff, but otherwise, welcome moving this piece forward.